# Validation of a UPLC-MS/MS Method for Quantifying Intracellular Olaparib Levels in Resistant Ovarian Cancer Cells

**DOI:** 10.3390/ph18121870

**Published:** 2025-12-08

**Authors:** Szymon W. Kmiecik, Jennifer Lewis, Jonas Schwickert, Henrik Breitenreicher, Martin R. Sprick, Jürgen Burhenne

**Affiliations:** 1Internal Medicine IX—Department of Clinical Pharmacology and Pharmacoepidemiology, Medical Faculty Heidelberg, Heidelberg University Hospital, Heidelberg University, Im Neuenheimer Feld 410, 69120 Heidelberg, Germany; 2Division of Stem Cells and Cancer, German Cancer Research Center (DKFZ), DKFZ-ZMBH Alliance, Im Neuenheimer Feld 280, 69120 Heidelberg, Germany; 3Heidelberg Institute for Stem Cell Technology and Experimental Medicine (HI-STEM gGmbH), Im Neuenheimer Feld 280, 69120 Heidelberg, Germany

**Keywords:** olaparib, UPLC-MS/MS, tandem mass spectrometry, cancer drug resistance, bioanalysis

## Abstract

**Background:** Ovarian cancer remains one of the leading causes of cancer-related mortality among women and constitutes a major unmet medical need. A common treatment-limiting factor for ovarian cancer patients is resistance to Poly(ADP-ribose) polymerase (PARP) inhibitors such as olaparib. Resistance mechanisms include restoration of functional homologous recombination repair, replication fork protection, *PARP1* mutations, and increased drug efflux or metabolism. Understanding these cellular and molecular mechanisms is essential for developing more effective therapeutic strategies and improving patient outcomes. **Methods:** In this study, patient-derived ovarian cancer cells (OC12) in which resistance to olaparib was induced by exposing the cells to increasing concentrations of the drug over multiple treatment cycles were investigated. To compare intracellular olaparib levels in sensitive and resistant cell lines, a UPLC-MS/MS method to quantify olaparib in the range of 1–300 ng/mL was developed. **Results:** The method was validated for selectivity, calibration curve performance, carryover, dilution integrity, precision, accuracy, matrix effect, and recovery in accordance with ICH M10 guidelines for bioanalytical method validation. Our findings revealed no significant difference in olaparib levels between resistant and sensitive OC12 cells, excluding the involvement of efflux transporters or enhanced metabolism of olaparib in the resistant OC12 ovarian cancer cells. **Conclusions:** These results shift the future focus toward pharmacodynamic factors as key drivers of olaparib resistance in OC12 cells. Taken together, the developed UPLC-MS/MS analytical method can be successfully applied to quantify intracellular olaparib levels and investigate the potential contribution of drug efflux mechanisms or increased metabolic activity in cells resistant to olaparib treatment.

## 1. Introduction

Ovarian cancer persists as a critical health concern, ranking as the fifth leading cause of cancer-related deaths in females in the United States, with a five-year overall survival rate of approximately 30% for advanced stages [1]. One of the most common genetic risk factors for ovarian cancer includes germline mutations in Breast Cancer gene 1 and 2 (*BRCA1* and *BRCA2*, or *BRCA1/2*). By age 70, women with *BRCA1* mutations have a 39% risk of developing ovarian cancer, while those with *BRCA2* mutations have an 11% risk [2]. These tumor suppressor genes play an important role in homologous recombination (HR)-mediated repair of double-stranded DNA breaks [3]. *BRCA1/2* mutations make cancer cells particularly vulnerable to inhibitors of Poly(ADP-ribose) polymerase protein (PARP), an enzyme that plays a role in sensing and repairing DNA damage. Therefore, PARP inhibitors (PARPis) are used in therapy for ovarian cancer patients with *BRCA1/2* mutations, and olaparib was the first PARPi approved for therapy [1].

The development of resistance to olaparib is a common treatment-limiting factor for patients with ovarian cancer. Understanding the molecular basis underlying cancer cells’ resistance to this therapy remains a major challenge. Several possible mechanisms by which cancer cells acquire resistance to olaparib have been identified. These include reactivating DNA repair machinery by secondary mutations in *BRCA1/2* that restores protein function, mutations in the *PARP1* gene, and upregulation of drug efflux transporters such as ATP Binding Cassette Subfamily B Member 1 (ABCB1), also known as Multidrug Resistance Protein 1 (MDR1) or P-glycoprotein (P-gp) [3,4]. In addition to the increased expression of drug efflux pumps, elevated drug metabolism by enzymes such as Cytochrome P450 3A (CYP3A) may also reduce intracellular olaparib concentration, potentially leading to drug resistance [5,6]. The immunosuppressive and typically immune “cold” microenvironment of ovarian cancer further impedes treatment effectiveness, underscoring the need for combinatorial therapeutic approaches that integrate immunotherapy with agents like olaparib to potentiate anti-tumor responses and overcome resistance mechanisms [7]. Emerging therapies under investigation for ovarian cancer include antibody-drug conjugates (ADCs) targeting folate receptor alpha (FRα) [8], anti-angiogenesis drugs, PI3K/AKT/mTOR signaling pathway inhibitors [9], recombinant arazyme [10], and targeting Rhophilin rho GTPase binding protein 1-antisense RNA 1 (RHPN1-AS1) [11]. These strategies offer potential for overcoming drug resistance and improving clinical outcomes in ovarian cancer.

In this study, resistance to olaparib was induced in patient-derived ovarian cancer cells (OC12) by exposing them to increasing concentrations of olaparib over multiple treatment rounds. An assessment of common resistance mechanisms revealed that the observed resistance was not caused by overexpression of the *ABCB1* or *ABCB4* efflux transporters. To further investigate the underlying mechanism behind the olaparib resistance in OC12 cells, an Ultra Performance Liquid Chromatography coupled with Tandem Mass Spectrometry (UPLC-MS/MS) analytical method was developed. The method allows measurement of the intracellular concentration of olaparib using triple quadrupole MS/MS with positive electrospray ionization (ESI) in the multiple reaction monitoring (MRM) mode. The olaparib mass transition of *m*/*z* 435.2 ⟶ 367.2 was used for quantification, and stable isotopically labeled ^2^H_4_-olaparib standard was used for signal normalization. The calibrated olaparib concentration range was 1 (Lower Limit of Quantification/LLOQ) to 300 ng/mL (Upper Limit of Quantification/ULOQ). The method was subsequently validated following ICH M10 guidelines with regard to cellular determinations [12]. Accuracy and precision were within the range of ±20% for LLOQ (1 ng/mL) and ±15% for quality control levels at 3, 50, and 225 ng/mL. This robust analytical method enabled reliable intracellular drug quantification, which is essential to differentiate whether reduced olaparib efficacy in resistant OC12 cells arises from insufficient intracellular drug levels (pharmacokinetic resistance) or from an altered cellular response despite adequate intracellular exposure (pharmacodynamic resistance) [13]. Employing this validated approach, the results showed no difference in olaparib concentrations between olaparib-sensitive and -resistant OC12 cell lines, suggesting that resistance is not due to pharmacokinetic mechanisms such as increased efflux or enhanced metabolism of the drug, but rather due to pharmacodynamic factors.

## 2. Results

### 2.1. Cell Culture Experiments

#### 2.1.1. Generation of PARPi (Olaparib) Resistance in Ovarian Cancer Cell Lines (OC12)

To generate PARPi-resistant cell lines from patient-derived olaparib-sensitive cell lines, cells were treated for several treatment rounds with increasing concentrations of olaparib. Cells were exposed to olaparib or DMSO as a vehicle control for four days, with culture medium exchange after 3 days. Subsequently, cells were given a drug holiday to recover and proliferate until they reached confluency (Figure 1A). Following this phase, the subsequent treatment cycle was repeated with an increased dose of olaparib (Figure 1B). Obtained resistant OC12 cell lines (R1, R2, R3) together with sensitive cell lines (S1, S2, S3), from which they had been derived, were tested for cell viability at increasing concentration of olaparib, demonstrating a huge increase in inhibitory concentration 50% (IC_50_) value for resistant OC12 cell lines as compared to olaparib-sensitive lines (Figure 1C and Appendix A). Single-proliferation curves for resistant and sensitive OC12 cell lines are indicated in Appendix A. One of the developed cell lines (R2) showed increased resistance to olaparib compared to the other two resistant lines, which affected the standard deviation of the calculated mean IC_50_ value. When this value was excluded, the difference in IC_50_ values for sensitive and resistant cell lines became significant (Appendix A).

#### 2.1.2. Upregulation of the Drug Efflux Transporter ABCB1 (P-glycoprotein) in OC12 Cells Resistant to Olaparib

The upregulation of ABCB1 (P-glycoprotein) is a well-known PARPi resistance mechanism [3]. It serves as a drug efflux transporter to reduce intracellular drug concentration and pump compounds outside of the cell [14]. Indeed, we found that ABCB1 was significantly upregulated in the resistant OC12 cell lines, with a 327-fold upregulation (R1) after treatment round 10, compared with olaparib-sensitive cells (Figure 2A). This suggests that ABCB1 upregulation could be associated with the acquisition of PARPi tolerance. To ascertain whether the olaparib resistance in the OC12 lines was directly linked to ABCB1 upregulation, CRISPR/Cas9-guided knockouts (KOs) of *ABCB1* in OC12 R1 and OC12 S1 were generated. Protein expression analysis in a Western Blot (WB) demonstrated that ABCB1 protein was abolished in the KO cells as compared to the non-targeting (NT) control of OC12 R1 (Figure 2B). To test whether *ABCB1*-deficient OC12 R1 cells could be re-sensitized to olaparib treatment, the cell viability assay comparing *ABCB1* wild type (WT) with *ABCB1* KO cells (Appendix A) was repeated. Even though the *ABCB1* KO markedly reduced the IC_50_ values of the resistant cells (from 26 μM to 6 μM), they were not sensitized to the IC_50_ values of the sensitive cells (0.24–0.58 µM). In addition, the IC_50_ values for sensitive cell lines were also reduced upon *ABCB1* KO (Appendix A). This allows us to speculate that ABCB1 upregulation was not the mechanism for resistance in the OC12 R1 cells. Since ABCB1 can have combined effects with ABCB4 to influence resistance to PARPis, a double KO of *ABCB1* + *ABCB4* (Appendix A) was performed. Due to the lack of suitable antibodies for WB analysis of ABCB4, the KO was confirmed with Sanger sequencing, resulting in a KO efficiency of 95% for OC12 R1 and 85% for S1. These numbers were comparable with the single KO efficacy of *ABCB1* in OC12 R1 (92%) and S1 (90%) (Appendix A), for which the findings were validated by WB analysis (Figure 2B).

Using the double KO, cell viability assays were performed, treating the resistant OC12 R1 and sensitive S1 *ABCB1/4* double KO as well as resistant OC12 R1 and sensitive S1 WT control cells with indicated concentrations of olaparib for five days. *ABCB1/4* double KO did not lead to a further decrease in IC_50_ value for the resistant OC12 R1 cells as compared to *ABCB1*-only KO (Appendix A), suggesting that ABCB1/4-driven drug efflux is not the driver of PARPi resistance in OC12 cells. Next, a cell viability assay using pamiparib, a potent and specific PARP1 and PARP2 inhibitor, was performed. Pamiparib is designed not to be a substrate for ATP Binding Cassette (ABC) transporters, offering a potential treatment option for tumors that have developed PARPi resistance by upregulating ABCB1 and ABCB4 [15]. The olaparib-resistant and -sensitive OC12 cells were treated with different concentrations of pamiparib for five days. In line with the *ABCB1* and *ABCB4* KO data, the resistant cells were also resistant to pamiparib with an IC_50_ value of 14.75 μM (Figure 2C,D).

Based on cell culture data, it is concluded that the upregulation of both drug efflux pumps, ABCB1 and ABCB4, has a small impact on the observed PARPi tolerance but is not responsible for the observed resistance. This conclusion is supported by the minor changes in IC_50_ values when knocking out *ABCB1* and *ABCB4*, as well as the resistance of olaparib-resistant cells to the treatment with pamiparib, which is not a substrate for ABC transporters.

To follow up on the possibility that drug transporters other than ABCB1 or ABCB4 are involved, or that olaparib is metabolized in resistant OC12 cells, a UPLC-MS/MS analytical method to quantify intracellular olaparib levels in resistant and sensitive OC12 cells was developed.

### 2.2. Mass Spectrometric Characteristics of Olaparib

Positive electrospray ionization of olaparib (C_24_H_23_FN_4_O_3_, 434.18 g/mol) yielded a singly charged species with the [M+H]^+^ signal at *m*/*z* 435.2 (*m*/*z* 439.2 for the internal standard (IS) ^2^H_4_-olaparib). The collision-induced dissociation (CID) fragmentation of olaparib with 18 eV resulted in four main daughter ions (Figure 3A). The loss of the cyclopropane ring leads to the formation of the 367.2 ion (the most intensive ion) and the 69.1 ion. Additionally, further cleavage of the piperazine ring generates the 324.1 ion, while cleavage of the amide bond adjacent to the piperazine ring yields the 281.1 ion (Figure 3B [16]). Because of the low background and the absence of interfering signals for the most abundant fragment at *m*/*z* 367.2, the corresponding mass transition *m*/*z* 435.2 ⟶ 367.2 was chosen for quantification using multiple reaction monitoring (MRM). To ensure that the IS accounted for variations in mass spectrometric detection, the corresponding mass transition of *m*/*z* 439.2 ⟶ 367.2 was monitored for the IS. MS/MS with MRM enables highly selective and sensitive quantification of small-molecule drugs by monitoring specific precursor-to-product ion transitions, effectively minimizing matrix interference and improving accuracy. In MRM mode, a triple quadrupole instrument isolates, fragments, and detects analyte-specific ions, which enhances signal-to-noise and facilitates reliable measurement even in complex biological samples [17].

### 2.3. Chromatographic Characteristics

Olaparib was chromatographically separated via UPLC using reversed-phase (RP) chromatography with a C18 column. Samples were loaded onto the column in a mobile phase containing a low concentration (~5%) of acetonitrile (ACN). An effective separation of potential interfering signals and optimal sample throughput was achieved by applying a rapid gradient from 5% to 95% ACN within 1.5 min. Figure 3C demonstrates the ACN gradient and a typical ion chromatogram for the selected transition of olaparib, revealing that the compound elutes at a point where the ACN concentration exceeds 50%.

### 2.4. Validation Results

The quantification of olaparib by the UPLC-MS/MS method, following extraction using protein precipitation, was performed in accordance with the relevant ICH M10 guidelines. Validation batches showed no significant interference in the blank cell samples, demonstrating the method’s selectivity. The average signal in blank cell samples equaled 1.1% for olaparib and 0.4% for IS (^2^H_4_-olaparib) of the corresponding average signal for LLOQ samples.

The response on the UPLC-MS/MS analytical platform was calculated as a ratio between integrated peak areas for the analyte (olaparib) and internal standard (^2^H_4_-olaparib). Exemplary chromatograms for calibration solutions are shown in Figure 4A (for olaparib) and Figure 4B (for ^2^H_4_-olaparib). The linearity of the calibration curves was confirmed by correlation coefficients (r^2^) exceeding 0.99 for all curves; weighted (1/x^2^) linear regression was used (Appendix A).

Carryover was analyzed by injecting an eluent sample after the highest olaparib calibration solution (300 ng/mL). The average signal in eluent samples equaled 2.7% for olaparib and 0.2% for IS (^2^H_4_-olaparib) of the corresponding average signal for LLOQ samples.

The inter-run and run-to-run (intra-run) accuracies of both assays were between 94.5 and 112.5% (100.4 and 108.9% at LLOQ) with a corresponding precision ≤10.9% (≤5.2% at LLOQ) (Table 1 and Table 2).

To ensure that the validation using p388 cells would also apply to OC12 cells, we performed the third validation batch using p388 cells for calibration samples but OC12 cells for QC samples. The obtained results demonstrate that OC12 QC samples align with the p388 cell calibration, indicating that the method can be applied to measure olaparib concentrations in OC12 cells. In addition, matrix effect and recovery were evaluated for both cell types used: p388 and OC12. IS (^2^H_4_-olaparib)-normalized matrix effect and recovery for QC levels at 3 and 225 ng/mL for p388 and OC12 cells were between 100.2% and 110.4% (Appendix A).

As the study samples for the first replicate were above the calibration range, therefore, for the second and third replicate, the samples were lysed in 50 µL 10% ammonia solution and then diluted five-fold with 10% ammonia solution. Subsequently, 50 µL of the diluted lysate was used for further analysis. Dilution integrity for five-fold dilution factors was evaluated at a QC-C concentration level showing 106.6% variation when comparing the IS-normalized response for diluted and undiluted samples (Appendix A).

Olaparib stability was thoroughly investigated in previous studies [18,19,20,21,22,23,24,25] and thereby was not explored in this work.

The developed method for olaparib quantification was validated with respect to selectivity, calibration curve performance, carryover, dilution integrity (five-fold dilution) precision, accuracy, matrix effect, and recovery. We demonstrate that the bioanalytical method complies with the recommended requirements of ICH M10 guidelines, underscoring the reliability of the olaparib bioanalysis.

### 2.5. Olaparib Concentration in Sensitive and Resistant OC12 Cell Lines

Using the validated bioanalytical method, the intracellular concentration of olaparib in resistant (R1) and sensitive (S1 and S2) OC12 cell lines was evaluated. Briefly, cells were incubated with medium containing 10 µM olaparib for 0.5, 4, 8, and 24 h, washed twice with ice-cold PBS buffer, lysed with 10% ammonium, and subsequently processed and subjected to UPLC-MS/MS analysis as described in the Methods section. The detected olaparib amount slightly increased over time in all tested cell lines, indicating olaparib accumulation. However, statistical analysis revealed no significant difference between the resistant and either of the two sensitive OC12 cell lines at any timepoint (Figure 5A). This lack of significant difference was further confirmed when normalizing olaparib amounts to one sensitive cell line (S1) within each timepoint (Figure 5B). One might also wonder if normalizing the samples to total protein concentration would influence the analysis; this is not the case. The data normalized to protein concentration and then to sensitive cell lines (S1) indicated no significant difference between resistant and sensitive cell lines (Appendix A). While our data’s variability might obscure minor differences in olaparib concentrations between resistant and sensitive cells, it is unlikely that such small variations in intracellular olaparib levels would substantially contribute to the observed drug tolerance in OC12 cells.

## 3. Discussion

While PARPis have demonstrated clinical efficacy in patients with HR-deficient tumors, long-term treatment success is often limited by the development of resistance. Several resistance mechanisms to PARPis have been identified, including restoration of HR proficiency through the reversion of *BRCA1/2* mutations, alterations in DNA-damage response pathways, and drug-related effects such as increased expression of drug efflux transporters (ABCB1/MDR1) or increased drug metabolism (reviewed in [26,27,28]).

Analysis of the olaparib-resistant OC12 cells did not reveal any mutational changes, leaving the underlying resistance mechanism elusive. Resistant OC12 cells demonstrate tolerance to PARPis, suggesting that the acquired resistance may be induced by metabolic or epigenetic changes initiated by the olaparib (drug-related effects). ABCB1/MDR1 overexpression has been identified as a mechanism of olaparib resistance in ovarian cancer cells [3]. However, this mechanism does not appear to explain resistance in OC12 cells. Despite the higher expression of *ABCB1* in resistant OC12 cells, its knockdown does not re-sensitize these cells to olaparib treatment. Moreover, even the combined knockout of *ABCB1* and *ABCB4* fails to fully restore olaparib sensitivity in OC12 cells. In addition, olaparib-resistant OC12 cells are also resistant to pamiparib, a PARP inhibitor that is not a substrate of ABC efflux pumps.

To investigate the potential involvement of additional efflux transporters or increased drug metabolism (e.g., by CYP3A enzymes) in resistant OC12 cells, a bioanalytical method for intracellular olaparib determination using the UPLC-MS/MS methodology was developed. The method was validated according to regulatory ICH M10 requirements. The established concentration range of quantification is from 1 ng/mL (LLOQ) to 300 ng/mL (ULOQ). Mass transitions *m*/*z* 435.2 ⟶ 367.2 (olaparib) and *m*/*z* 439.2 ⟶ 367.2 (IS; ^2^H_4_-olaparib) were used in MRM. To our knowledge, only one study has reported a method for olaparib detection with higher sensitivity (LLOQ at 0.1 ng/mL) than our method (LLOQ at 1 ng/mL) [23]. In contrast, seven other studies [18,19,20,21,22,24,25] were unable to detect olaparib at concentration levels below 10 ng/mL.

No significant difference in the amount of olaparib between resistant and sensitive OC12 cells was detected using the developed UPLC-MS/MS method. These findings strongly suggest that the acquired resistance in OC12 cells is not due to a lower intracellular concentration of olaparib in resistant cells, further ruling out a role of efflux transporters in the acquired olaparib tolerance or increased olaparib metabolism in resistant OC12 cells. Other mechanisms of resistance to PARPis have been described, such as restoration of HR activity, replication fork protection, and mutations in *PARP1* or Poly(ADP-ribose) glycohydrolase (*PARG*) that render PARPis ineffective [29,30]. These findings shift the future focus toward pharmacodynamic factors as mechanisms driving olaparib resistance in OC12 cells.

Olaparib is widely used in therapy and exhibits broad tissue distribution with moderate tumor accumulation and limited brain penetration due to active efflux transporters [31,32]. Efflux pumps such as ABCB1 can actively export olaparib from cells, reducing its intracellular concentration and contributing to drug resistance. The developed analytical method for quantifying olaparib can be utilized to study the impact of these efflux pumps on olaparib accumulation and to better understand resistance mechanisms. Furthermore, the assay could be adapted and cross-validated for quantifying olaparib in other biological matrices, such as human or mouse plasma, to assess pharmacokinetic parameters. This quantification protocol could also be employed to investigate olaparib penetration across intact and leaky (tumor-compromised) blood-brain barriers, as well as its plasma protein binding.

## 4. Material and Methods

### 4.1. Cell Culture

Suspension p388 cells (provided by Dr. D. Ballinari, Pharmacia & Upjohn, Milano, Italy) were cultivated in RPMI 1640 Medium (PAN-Biotech, Aidenbach, Germany; #P04-16500) supplemented with 10% Fetal Bovine Serum (FBS) (PAN-Biotech, Aidenbach, Germany; #P30-3306), 2 mM Glutamine (Sigma-Aldrich, St. Louis, MO, USA; #G7513), 100 U Penicillin/100 µg Streptomycin (Sigma-Aldrich, St. Louis, MO, USA; #P4333), and 100 µM 2-mercaptoethanol (Sigma-Aldrich, St. Louis, MO, USA; #M3148) at 37 °C and 5% CO_2_. Cells were harvested (450× *g*, 5 min, 4 °C), washed once with ice cold Phosphate-buffered saline (PBS), aliquoted (10^6^ cells per aliquot), and harvested (1000× *g*, 5 min, 4 °C). The cell pellets were subsequently frozen and stored at −20 °C.

OC12 cells (adenocarcinoma of the ovary, ascites origin), derived from a patient treated with cyclophosphamide, adriamycin, and cisplatin [33], are a serum-free culture-adapted sub-line of OVCAR-3 human ovarian carcinoma (Sigma-Aldrich, St. Louis, MO, USA; #SCC257), which is refractory to cisplatin, but remains sensitive to a number of chemotherapeutic drugs [34]. They were maintained in Ovarian TumorMACS™ Medium (Milteny Biotec, Bergisch Gladbach, Germany; #130-119-483) in an adherent cell culture at 37 °C and 5% CO_2_. For passaging, cells were detached using Accutase (Thermo Fisher, Waltham, MA, USA; #A11105) and were collected with COBG Medium (CO_2_-independent medium (Thermo Fisher, Waltham, MA, USA; #18045088), supplemented with 1% Bovine Serum Albumin (BSA) and 2 mM L-glutamine). Cells were centrifuged (300× *g*, 5 min, room temperature/RT), the cell pellet resuspended in TumorMACS™ Medium, and the cells were seeded on a new Corning^®^ Primaria^TM^ culture vessel (Sigma, St. Louis, MO, USA; #CLS353810) in a 1:20 ratio.

#### 4.1.1. Generation of Olaparib-Resistant Cell Lines

To generate the olaparib-resistant cells, the basal OC12 cell line was cultured in six separate flasks, with three flasks treated with olaparib (Selleckchem, Houston, TX, USA; #S1060) and the remaining three treated with dimethyl sulfoxide (DMSO) (Sigma, St. Louis, MO, USA; #D2650) as a vehicle control. Each flask was treated for 4 days with the specified concentrations of olaparib or DMSO, with medium replacement after 48 h, followed by a drug holiday period until the cells reached 90% confluence. This treatment cycle was repeated for 11 rounds, with progressively increasing doses of olaparib (based on [35]).

To analyze the amount of olaparib in OC12 cells, cells were seeded on day 0 (1 × 10^6^ cells/well) in a 6-well Corning^®^ Primaria^TM^ plate (Sigma, St. Louis, MO, USA; #CLS353846). Twenty-four-hour post-seeding cells were treated with 10 µM olaparib for different timepoints. After incubation, cells were placed on ice and washed two times with ice cold PBS. Cells were lysed in the well with 50 µL of 10% ammonium hydroxide and transferred to a 1.5 mL Eppendorf tube. Total protein concentration in lysates was measured using a Qubit Fluorometer (Thermo Fisher, Waltham, MA, USA; #Q33238) and Qubit Broad Range Protein Assay Kit (Thermo Fisher, Waltham, MA, USA; #Q33211). Cell pellets were frozen at −80 °C for further analysis.

#### 4.1.2. Western Blot Analysis

Whole-cell lysates of OC12 cells were prepared using RIPA buffer, 1 mM EDTA, 1 mM AEBSF, and 1 mM Halt Protease/Phosphatase Inhibitor cocktail (Thermo Fisher, Waltham, MA, USA; #78429). Protein lysates were resolved on 4–20% Criterion™ TGX Stain-Free™ protein gels (BioRad, Hercules, CA, USA; #5671094) with TGS running buffer (BioRad, Hercules, CA, USA; #1610732) and plotted on a PVDF membrane (BioRad, Hercules, CA, USA; #1704157). The membrane was blocked for 1 h at RT in TBS containing 0.1% Tween-20 with 5% (*w*/*v*) BSA (blocking solution). Primary antibodies (anti-ABCB1, rabbit IgG, 1:1000, Cell Signaling Technology, Danvers, MA, USA; #13978; anti-a-tubulin, mouse IgG, 1:1000, Cell Signaling Technology, Danvers, MA, USA; #3873) were incubated over night at 4 °C in blocking solution. Secondary HRP-coupled antibodies (anti-rabbit IgG, 1:10,000, Cell Signaling Technology, Danvers, MA, USA; #7074; anti-mouse IgG1, 1:10,000, Biozol, Hamburg, Germany; #1071-05) were diluted in blocking solution for 1 h at RT. The membrane was washed in 0.1% TBS-Tween, and immunocomplexes were detected using an ECL chemiluminescence kit (BioRad, Hercules, CA, USA; #1705060) [6,36].

#### 4.1.3. Real-Time Quantitative PCR

Total RNA was extracted using an miRNeasy mini kit (Qiagen, Hilden, Germany; #217004) and reverse-transcribed using a high-capacity cDNA reverse transcription kit (Applied Biosystems, Waltham, MA, USA; #4374966). cDNA corresponding to 10 ng of the starting RNA was used for relative RNA quantification (qRT-PCR). TaqMan probes (Thermo Fisher, Waltham, MA, USA; #4331182) for *ABCB1* (Hs00184500_m1)), PPIA (Hs99999904_m1), TBP (Hs00427620_m1), and POLR2A (Hs00172187_m1) were used to acquire expression data with the VIIA 7 Real-Time PCR System (Thermo Fisher, Waltham, MA, USA). The QuantStudio™ Design and Analysis software (v. 1.4.3, Thermo Fisher, Waltham, MA, USA) was used for data acquisition and analysis [6,37].

#### 4.1.4. Cell Viability Assay

For determination of relative cell viability, serial dilutions of olaparib were screened in quadruplicate. In brief, 1000 cells per well were seeded in 96-well plates 24 h prior to the addition of the individual compounds. After incubation for 5 days (medium was exchanged after 72 h with medium containing fresh drug), cell viability was assessed using CellTiterBlue (Promega, Madison, WI, USA; #G8080) following manufacturer’s instructions. Vehicle (DMSO) was used as a negative control. Treatment with 20 μM staurosporine (Selleckchem, Houston, TX, USA; #S1421) was used as a positive control. Responses were normalized to DMSO- and staurosporine-treated controls. Relative cell viability curves were plotted using GraphPad Prism (Version 10.5.0, GraphPad Software, San Diego, CA, USA).

#### 4.1.5. CRISPR-Cas9-Mediated Knockout

The crRNA targeting *ABCB1* Exon 3 (5′ GATCTTGAAGGGGACCGCAA 3′) was designed and ordered from Integrated DNA Technologies (IDT, Coralville, IA, USA). Electroporation was performed using a NEPA21 electroporator (Nepagene, Ichikawa, Japan) according to the manufacturer’s protocol. In brief, to generate the guide RNA (gRNA) duplex, the gene-specific Alt-R CRISPR-Cas9 crRNA (200 μM, IDT, Coralville, IA, USA) was combined with Alt-R CRISPR-Cas9 tracrRNA (200 μM, IDT, Coralville, IA, USA; #1072533) in an equimolar ratio, heated to 95 °C for 5 min, and then allowed to gradually cool to room temperature to form the crRNA duplex. The gRNA duplex was then assembled with Cas9 protein by mixing 4 µL of the duplex with 5 µL of Cas9 Nuclease (61 μM, IDT, Coralville, IA, USA; #1081061) and incubating it at room temperature for 20 min to form the ribonucleoprotein (RNP) complex. The RNP complex was prepared for electroporation by adding 16 µL of Opti-MEM™ (Gibco, Billings, MT, USA; #10149832) to the 9 µL RNP mixture. Cells were resuspended in 100 µL of Opti-MEM™ and then combined with 10 µL of the RNP complex. The cell/RNP suspension was transferred into an electroporation cuvette (2 mm gap), gently mixed by tapping, and subjected to electroporation using the NEPA21 electroporator (Nepagene, Ichikawa, Japan) under the following parameters: Poring Pulse (125 V, 5 ms (length), 50 ms (interval), 2 pulses, 10% D. Rate); Transfer Pulse (20 V, 50 ms (length), 50 ms (interval), 5 pulses, 40% D. Rate).

### 4.2. Analytical Development: Drugs, Chemicals, and Solvents

Olaparib was obtained from Biozol GMbH (Hamburg, Germany) supplied by RayBiotech Life (Norcross, GA, USA; #331-10930). Stably isotopically labeled (SIL) internal standard (IS) ^2^H_4_-olaparib was obtained from Biozol GMbH (Hamburg, Germany) supplied by Toronto Research Chemicals (Toronto, ON, Canada; #TRC-0514502). Arium^®^ Mini System (Sartorius, Göttingen, Germany) was used for production of ultra-pure water. Ammonium hydroxide solution was obtained from Sigma-Aldrich (St. Louis, MO, USA; #338818-100ml). Acetonitrile (ACN) and formic acid (FA) of the highest available purity were purchased from Biosolve (Valkenswaard, The Netherlands).

### 4.3. Preparation of Standard Solutions

To prepare stock solutions, independent weighings of olaparib (10.06 and 3.75 mg) were used. Olaparib was dissolved in 10 mL (to 1.006 mg/mL for calibration solutions) and 2 mL (to 1.875 mg/mL for quality control (QC) solutions) in ACN/H_2_O (1/1, *v*/*v*) using volumetric flasks. Spike solutions for the preparation of calibration samples were generated from the 10-times-diluted stock solution (1.006 mg/mL) at concentrations of 1, 3, 10, 30, 50, 100, and 300 ng/mL in ACN/H_2_O (5/95 + 0.1% FA, *v*/*v*). Spike solutions for preparation of QC samples were prepared from 100-times diluted stock solution (1.875 mg/mL) at 1 (LLOQ), 3 (QC-A), 50 (QC-B), and 225 (QC-C) ng/mL. The internal standard (IS) spike solution was prepared by 10^6^-dilution of the 1 mg/mL stock solution in ACN/H_2_O (5/95 + 0.1% FA, *v*/*v*) to yield 1 ng/mL. Solutions were kept at −20 °C.

### 4.4. Preparation of Cell Samples for UPLC-MS/MS Measurements

Calibration and QC cell samples were generated in 1.5 mL reaction tubes. An amount of 10^6^ p388 or OC12 cells were pelleted and lysed in 50 µL of 10% ammonia solution, and subsequently, 25 µL of IS and 25 µL of the respective calibration or QC spike solution were added. For blank and zero calibration samples, 25 µL of ACN/H_2_O (5/95 + 0.1% FA, *v*/*v*) was used for volume compensation; in addition, the blank sample was spiked with 25 µL of ACN/H_2_O (5/95 + 0.1% FA, *v*/*v*) instead of IS for volume compensation. Study OC12 cell samples (50 µL) were also spiked with 25 µL of IS solution and 25 µL of ACN/H_2_O (5/95 + 0.1% FA, *v*/*v*) for volume compensation.

Cell samples were depleted of proteins by the addition of 300 µL of ACN and subsequent centrifugation (16,200× *g*, 5 min, RT). Obtained extracts were transferred to a 1 mL 96-well collection plate (Waters, Eschborn, Germany, #186002481). The extracts were then blowdown-evaporated with heated nitrogen (40 °C) applied for 2 × 15 min at 4 psi with a blowdown evaporator (Ultravap^®^, Porvair Sciences, Wrexham, Wales, UK). Next, 175 µL of ACN/H_2_O (5/95 + 0.1% FA, *v*/*v*) was added to each well, plates were sealed, and sample extracts were vortex-mixed.

### 4.5. Instrumental Analysis Parameters

UPLC-MS/MS measurements were performed using an Acquity Classic UPLC^®^ (Waters, Milford, MA, USA) coupled with a triple-stage quadrupole mass spectrometer (Waters Xevo TQ-XS with Z-spray electrospray ionization (ESI) source). Capillary voltage was manually optimized. The remaining mass spectrometric parameters were tuned with the integrated IntelliStart procedures of the MassLynx system software (V4.2, Waters, Milford, MA, USA). Argon was used for collision-induced dissociation (CID) in positive-ion multiple-reaction monitoring (MRM) mode measurements. A summary of the mass spectrometric characteristics is shown in Table 3. For chromatographic separation, water including 5% of ACN and 0.1% of FA (mobile phase 1/MP1) and ACN including 0.1% FA (mobile phase 2/MP2) were used. Separation was carried out using an AQUITY UPLC^®^ Peptide BEH C18 Column (Waters, Milford, MA, USA; #186003685) maintained at 40 °C with a flow rate of 0.5 mL/min. A sample injection volume of 20 µL was used. After 0.5 min of column equilibration, a 5% to 95% MP2 gradient in 1.5 min was used for separation. Following the separation gradient, the column was washed with 95% MP2 for 0.5 min before returning to its initial condition in 0.5 min, which was maintained for 0.5 min during the preparation of the following injection by the Sample Manager. A total sample cycle time of 3.5 min was obtained.

### 4.6. Validation of the Analytical Method

Validation parameters were defined according to the pertinent recommendations of the Food and Drug Administration (FDA) and European Medicines Agency (EMA) summarized in the International Council for Harmonization (ICH) M10 guidelines [12]. Validation was performed in regard to accuracy and precision within-run and run-to-run in three validation runs. Each run contained blank p388 cells, one zero, and seven non-zero calibration concentrations, each in double determination, as well as six determinations of four QC concentrations (LLOQ, low QC at three times the LLOQ, mid QC at 16.6% of the ULOQ, and high QC at 75% of the ULOQ). Please note that for quality control samples in the last validation batch, OC12 cells were used. Accuracy was calculated from the mean determined concentration of QC samples as the percent of the nominal value. The respective acceptance limits are within ±15% of the nominal value (100%) with the exception of the LLOQ, which must lie within ±20%. Precision was evaluated from the standard deviation of measured QC samples as the percent of the mean determined concentration. Required limits are ≤15% in general and ≤20% at the LLOQ.

Selectivity was evaluated in three validation batches of blank p388 cells for absence of interfering peaks at the analyte retention time (peak area ≤20% of LLOQ and ≤5% of IS).

Carryover was accessed by running an eluent (ACN/H_2_O, 5/95 + 0.1% FA, *v*/*v*) sample after the highest calibration sample (300 ng/mL) and integrating olaparib peaks at the eluent sample (peak area ≤20% of LLOQ and ≤5% of IS).

Extraction recovery rates from p388 and OC12 cells were assessed from QC-A (3 ng/mL) and QC-C (225 ng/mL) samples in an at least four-fold determination as the ratio of response in regard to blank cells spiked after extraction (representing 100% analyte amount in identical matrix) expressed in percent (Appendix A).

Matrix effects were evaluated in an at least four-fold determination for QC-A (3 ng/mL) and QC-C (225 ng/mL) samples via the comparison of the response of blank cell samples spiked after extraction with the respective response of matrix-free extract solvent spiked with the identical amount of olaparib (Appendix A).

Dilution integrity was evaluated by spiking 10^6^ p388 cells lysed with 10% ammonia solution with olaparib (end volume 50 µL) to achieve five-fold higher content of olaparib as in the corresponding QC-C sample. The resulting lysate was then diluted five-fold with 10% ammonia solution, and 50 µL of the diluted lysate was further analyzed (Appendix A).

The method development and validation workflow is summarized in Appendix A.

### 4.7. Calculations, Statistical Methods, Figures Preparation

Calibration curves were calculated with weighted linear regressions (1/x^2^) from the response (peak area ratios of the analyte and IS) of calibration samples with TargetLynx software (V4.2, Waters, Milford, MA, USA). Remaining calculations were performed with Microsoft Office Excel (Version 2108, Microsoft Corporation, Redmond, WA, USA). GraphPad Prism (Version 10.5.0, GraphPad Software, San Diego, CA, USA) software was used to generate graphs and to perform statistical analyzes. The olaparib structure was drawn in ChemSketch (Version 2020.1.2, ADC/Labs, Toronto, ON, Canada). Figures were assembled in Inkscape (v.1.3.2, The Inkscape Project).

## 5. Conclusions

Accurate quantification of small-molecule drugs using UPLC-MS/MS enables detailed investigation of cellular drug processing, supporting comprehensive pharmacokinetic and pharmacodynamic studies [38]. The sensitive and precise method developed in this work allows measurement of intracellular olaparib concentrations, providing insights into resistance mechanisms in OC12 ovarian cancer cells, such as increased drug efflux and enhanced metabolism. This robust and reproducible technique is suitable for preclinical and clinical research and serves as a valuable analytical tool to understand olaparib resistance and guide the development of improved cancer therapies. It can also be used to explore resistance mechanisms mediated by pharmacodynamic factors. Future studies may expand the application of this method to other PARP inhibitors and tumor types to further elucidate resistance pathways. Integrating this approach into clinical monitoring, for instance through olaparib quantification in patient biopsies, could help personalize treatment strategies and improve patient outcomes.

## Figures and Tables

**Figure 1 pharmaceuticals-18-01870-f001:**
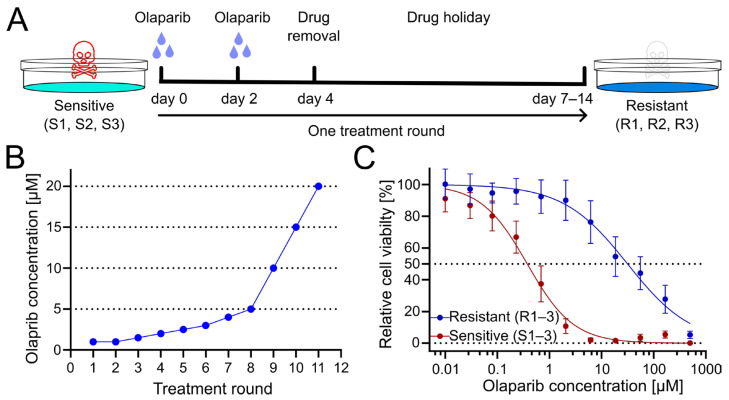
Treatment regimen to generate olaparib-resistant OC12 ovarian cancer cells. (**A**) Schematic representation of the olaparib treatment regimen, consisting of 2 + 2 days of treatment followed by a drug-free interval (drug holiday) to allow cells to recover. Three olaparib-resistant cell lines (R1, R2, R3) were generated from three olaparib-sensitive cell lines (S1, S2, S3). (**B**) The olaparib concentration at each treatment round. (**C**) Cell viability assay results for pool data for three sensitive (S1–3) and three resistant (R1–3) OC12 cell lines after the 10th treatment round. Cells were treated for 5 days with specified concentrations of olaparib, and relative cell viability was normalized to the corresponding DMSO control; mean value ± SD is shown; *n* ≥ 11; data were fitted in GraphPad Prism with a normalized response–variable slope model. Dotted lines at 0% and 50% relative cell viability are shown to improve readability.

**Figure 2 pharmaceuticals-18-01870-f002:**
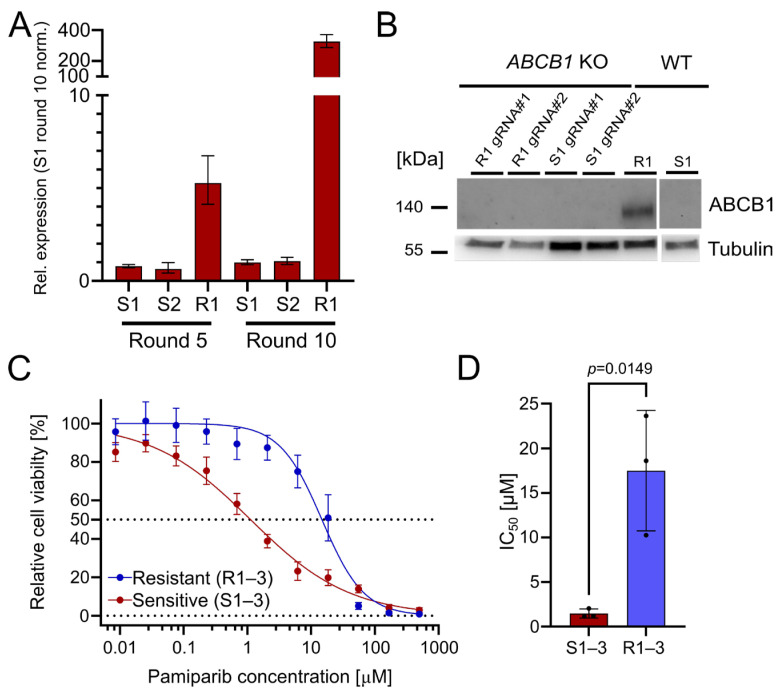
The drug efflux pumps ABCB1 and ABCB4 are upregulated in OC12 R cells. OC12 cells are also resistant to pamiparib, which is not a substrate of ABCB1. (**A**) *ABCB1* expression was assessed through qRT-PCR analysis in resistant and sensitive OC12 cells at treatment rounds 5 and 10; mean value ± propagated SD is shown. (**B**) Western Blot analysis of ABCB1 protein expression in OC12 cells with CRISPR-Cas9 KO and WT. (**C**) Cell viability assay results for pool data for three sensitive (S1–3) and three resistant (R1–3) OC12 cell lines. Cells were treated for 5 days with specified concentrations of pamiparib, and relative cell viability was normalized to the corresponding DMSO control; mean value ± SD is shown; *n* ≥ 11; data were fitted in GraphPad Prism with a normalized response–variable slope model. Dotted lines at 0% and 50% relative cell viability are shown to improve readability. (**D**) IC_50_ values calculated from 2C for sensitive (S1–3) and resistant (R1–3) OC12 cell lines treated with pamiparib; *n* = 3; mean value ± SD is shown; an unpaired T-test was used to evaluate statistical significance (*p* value = 0.0149).

**Figure 3 pharmaceuticals-18-01870-f003:**
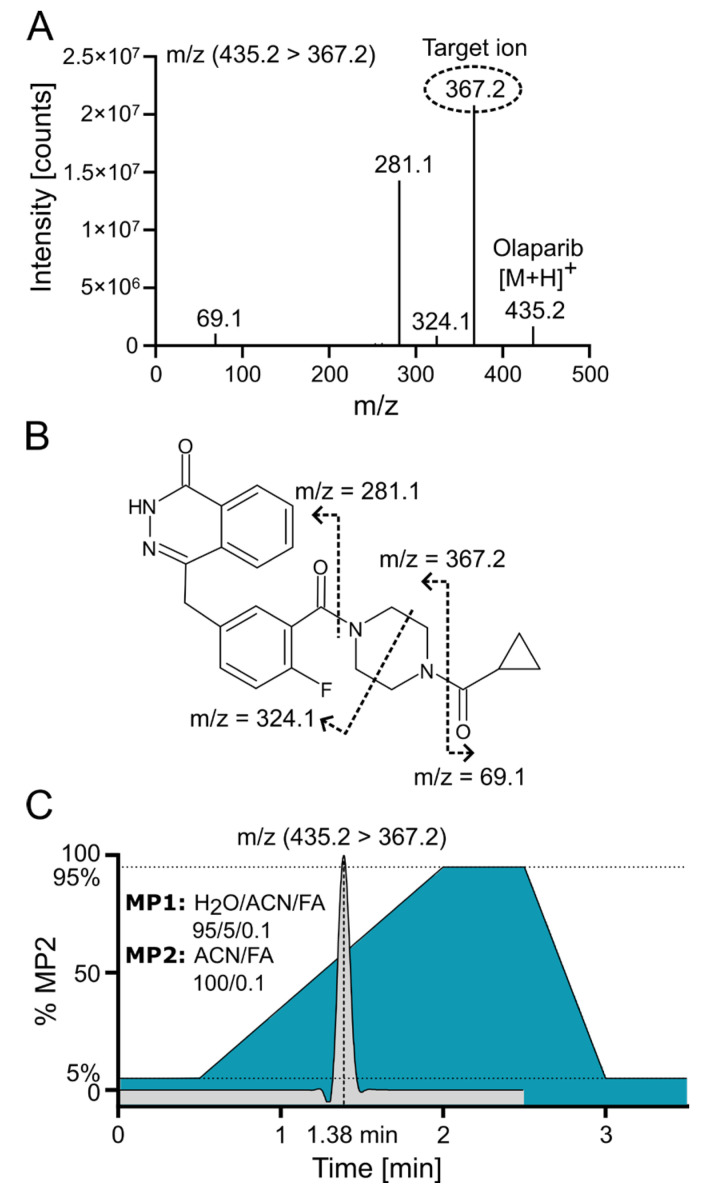
Olaparib fragmentation pattern and chromatographic separation. (**A**) Olaparib MS/MS fragmentation spectrum at 18 eV with indicated target/daughter ion used for quantification. (**B**) Predicted fragmentation pattern of olaparib [16]. (**C**) ACN gradient with olaparib elution profile from C18 reversed-phase (RP) column; FA—formic acid; MP—mobile phase.

**Figure 4 pharmaceuticals-18-01870-f004:**
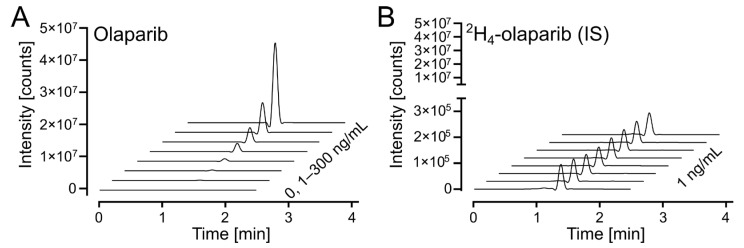
Olaparib calibration. (**A**) Waterfall plot of olaparib, *m*/*z* (435.2 > 367.2), in the calibrated range (1–300 ng/mL). (**B**) Waterfall plot of ^2^H_4_-olaparib (internal standard at 1 ng/mL), *m*/*z* (439.2 > 367.2), for olaparib calibration samples in 4A.

**Figure 5 pharmaceuticals-18-01870-f005:**
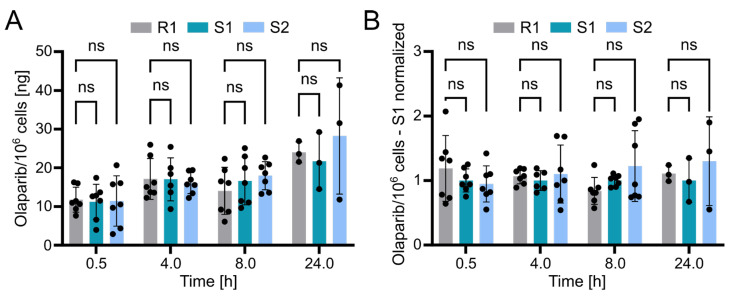
Olaparib detection in sensitive and resistant OC12 cells. (**A**) Amount of olaparib in OC12 cells measured in ng per 10^6^ cells. Single datapoints represent technical replicates from 3 independent experiments. Mean value ± SD is shown; *n* ≥ 6 for timepoints 0.5, 4, and 8 h; *n* = 3 for 24 h timepoint; 2-way ANOVA was used for statistical analysis; ns: not significant. The 24 h timepoint was analyzed in only one experiment. (**B**) Amount of olaparib in OC12 cells measured in ng per 10^6^ cells, normalized to olaparib amount detected in olaparib-sensitive (S1) sample. Normalization was performed within each timepoint. Single datapoints represent technical replicates from 3 independent experiments. Mean value ± SD is shown; *n* ≥ 6 for timepoints 0.5, 4, and 8 h; *n* = 3 for 24 h timepoint; 2-way ANOVA was used for statistical analysis; ns: not significant. The 24 h timepoint was analyzed in only one experiment.

**Table 1 pharmaceuticals-18-01870-t001:** Accuracy data of the cell validation.

QC Level	Nominal Concentration [ng/mL]	Accuracy [%]			
		Within-Batch			Batch-to-Batch
		Batch #1 p388	Batch #2 p388	Batch #3 OC12	
LLOQ	1	108.9%	100.4%	107.4%	105.6%
QC-A	3	106.7%	94.5%	108.3%	103.2%
QC-B	50	102.8%	97.0%	112.5%	104.1%
QC-C	225	97.0%	100.9%	110.5%	102.8%

Ratio between mean and nominal value is presented; LLOQ: lower limit of quantification, QC: quality control; *n* = 6 replicates; #: number.

**Table 2 pharmaceuticals-18-01870-t002:** Precision data of the cell validation.

QC Level	Nominal Concentration [ng/mL]	Precision [%]			
		Within-Batch			Batch-to-Batch
		Batch #1 p388	Batch #2 p388	Batch #3 OC12	
LLOQ	1	1.9%	4.0%	5.2%	5.2%
QC-A	3	2.6%	3.6%	6.2%	7.5%
QC-B	50	2.1%	2.5%	3.3%	6.8%
QC-C	225	2.8%	10.9%	4.7%	8.7%

Percentage coefficient of variation (%CV) is presented; LLOQ: lower limit of quantification, QC: quality control; *n* = 6 replicates; #: number.

**Table 3 pharmaceuticals-18-01870-t003:** Optimized MS/MS parameters for the detection of olaparib using heated ESI and MRM in the positive-ion mode.

Parameter	
Capillary voltage [kV]	3.0
Cone voltage [V]	20
Source temperature [°C]	150
Desolvation temperature [°C]	600
Cone gas (N_2_) flow [L/h]	150
Desolvation gas (N_2_) flow [L/h]	1000
Olaparib mass transition MRM [*m*/*z*]	435.2 ⟶ 367.2
^2^H_4_-olaparib (SIL-IS) mass transition MRM [*m*/*z*]	439.2 ⟶ 367.2
Collision gas (Ar) flow [mL/min]	0.15
Collision energy [V]	14
Dwell time [ms]	63

ESI: electrospray ionization; MRM: multiple reaction monitoring; SIL-IS: stable isotopically labeled internal standard.

## Data Availability

The original contributions presented in this study are included in the article/Appendix A. Further inquiries can be directed to the corresponding author.

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
