# Peer review of "Validation of a UPLC-MS/MS Method for Quantifying Intracellular Olaparib Levels in Resistant Ovarian Cancer Cells"

_pharmaceuticals, 2025, doi:10.3390/ph18121870_

Round 1

Reviewer 1 Report

Comments and Suggestions for Authors
  • The manuscript should be written predominantly in the passive voice to maintain a formal and objective scientific tone.

  • The sources of materials used in the study should be reported in the following standardized format: (Company, City, Country).

  • Several sections of the methodology lack appropriate references. Specifically, citations should be added for the following procedures:

    • Generation of the olaparib-resistant cell line

    • Western blot analysis

    • Polymerase Chain Reaction (PCR)

  • The software used in the study should be properly cited, including its name, version, and manufacturer/source.

  • The design of the analytical development should be described in more detail, specifying the different parameters that were optimized or validated.

  • The role of MS/MS in the analysis should be clearly explained, and the specific MS/MS parameters used in the study should be reported.

  • In the cell line experiments, there appears to be no clear correlation between apoptosis and necrosis. This relationship should be analyzed and discussed to strengthen the biological interpretation.

  • The manuscript currently lacks a conclusion section. A concise conclusion summarizing the key findings, implications, and recommendations should be added.

  • The reference list requires updating to include more recent and relevant studies that reflect the current state of research in this field.

Author Response

Please see the attached pdf to Reviewer #1

Reviewer 2 Report

Comments and Suggestions for Authors

Dear Authors, I have some small suggestions for the manuscript of your paper:

Manuscript should have numerated lines of text for easier reference.

It is better to use square brackets for literature reference [] instead of ().

All abbreviations in the text must be explained at the location where they are first mentioned (for example, BRCA1 and BRCA2 (BRCA1/2)).

Figure 1. and Figure 2. (same comment for Figure 3. and 4.) - maybe images A, B, C, D should each be a new figure, and adequate space should be provided for each chart (they seem too condensed).

After Discussion, there should be a separated chapter, named Conclusion.

Author Response

Please see the attached pdf to Reviewer #2

Reviewer 3 Report

Comments and Suggestions for Authors

This study presents a investigation into olaparib resistance mechanisms in ovarian cancer, emphasizing the development and validation of a UPLC–MS/MS bioanalytical method for quantifying intracellular olaparib levels. The focus on excluding efflux and metabolic mechanisms contributes meaningfully to understanding resistance pathways. However, to enhance the scientific impact and clarity, it required revision as per following comments:

  1. Abstract: The background section effectively establishes the clinical relevance of olaparib resistance but could benefit from one sentence contextualizing known resistance mechanisms (e.g., DNA repair reactivation, PARP1 mutations).
  2. Abstract: The conclusion should explicitly connect the analytical results to biological significance, linking the absence of efflux involvement to potential DNA repair–mediated resistance mechanisms.
  3. The rationale could be strengthened by briefly explaining why intracellular drug quantification is essential for distinguishing between pharmacokinetic versus pharmacodynamic resistance mechanisms.
  4. Including a statement such as “Resistance may originate from altered target engagement or DNA repair reprogramming rather than drug efflux or metabolism” would better integrate biological reasoning.
  5. The quantification range (1–300 ng/mL) is appropriate, but indicating the limit of detection (LOD) and limit of quantification (LOQ) would improve the technical completeness.
  6. The interpretation that efflux or metabolic differences are not major contributors is sound but should be contextualized with possible compensatory mechanisms (e.g., restoration of homologous recombination or replication fork protection).
  7. It would strengthen the discussion if the authors noted that such findings redirect future focus toward DNA repair pathway modulation as a primary driver of resistance.
  8. The conclusion should explicitly connect the validated method to future biological exploration, e.g., “This validated approach can be extended to investigate resistance mechanisms involving DNA repair reactivation or altered PARP trapping efficiency.” A short statement on potential clinical relevance, such as the use of intracellular olaparib quantification in patient biopsies, would significantly enhance the translational outlook.
  9. The introduction should supplement the current research and enrich the content of the paper. For example: https://doi.org/10.14744/ejmo.2023.32682, https://doi.org/10.1111/imm.13793, https://doi.org/10.1080/21655979.2021.2002494

Author Response

Please see the attached pdf to Reviewer #3

Reviewer 4 Report

Comments and Suggestions for Authors

Dear Authors,

I have carefully reviewed the paper by Kmiecik et al., titled "Validation of a UPLC-MS/MS method for quantifying intracellular olaparib levels in resistant ovarian cancer cells," and I am pleased to announce that I have accepted it for publication in pharmaceuticals Journal in the present form.

My decision is based on the following observations and considerations.

The manuscript presents the development and full validation of a UPLC-MS/MS method for determining intracellular olaparib concentrations in ovarian cancer cells (OC12 lines) and its application to compare drug levels in sensitive and resistant cells. The authors demonstrate that, despite the induction of resistance, olaparib levels in resistant and sensitive cells are not statistically significantly different, suggesting that the resistance is not due to impaired drug transport or metabolism.

Strengths of the work:

Carefully conducted analytical validation - the method meets ICH M10 requirements for selectivity, precision, accuracy, matrix effect, recovery, calibration range, and dilution integrity.

Wide range of method applications - the work highlights the adaptability of the protocol to other biological matrices and the potential for pharmacokinetic evaluation.

A very detailed description of the instrumentation and procedures - full LC-MS/MS parameters (capillary, gradient, MRM transitions, retention times) are provided in a manner that allows for method replication.

Significant biological contribution - the authors combine bioanalytical validation with biological analysis, providing insights into resistance mechanisms in OC12 cells.

Clear presentation of results - numerous graphs and chromatograms and concise interpretation of the data.

kind regards,

Author Response

Please see the attached pdf to Reviewer #4

Round 2

Reviewer 1 Report

Comments and Suggestions for Authors

Thanks for doing modifications